# INO80 exchanges H2A.Z for H2A by translocating on DNA proximal to histone dimers

Sandipan Brahma[1,2,3,*], Maheshi I. Udugama[3,*,†], Jongseong Kim[4,*,†], Arjan Hada[1,2,3], Saurabh K. Bhardwaj[1,2,3,†], Solomon G. Hailu[1,2,3], Tae-Hee Lee[4] & Blaine Bartholomew[1,2]

ATP-dependent chromatin remodellers modulate nucleosome dynamics by mobilizing or disassembling nucleosomes, as well as altering nucleosome composition. These chromatin remodellers generally function by translocating along nucleosomal DNA at the H3–H4 interface of nucleosomes. Here we show that, unlike other remodellers, INO80 translocates along DNA at the H2A–H2B interface of nucleosomes and persistently displaces DNA from the surface of H2A–H2B. DNA translocation and DNA torsional strain created near the entry site of nucleosomes by INO80 promotes both the mobilization of nucleosomes and the selective exchange of H2A.Z–H2B dimers out of nucleosomes and replacement by H2A–H2B dimers without any additional histone chaperones. We find that INO80 translocates and mobilizes H2A.Z-containing nucleosomes more efficiently than those containing H2A, partially accounting for the preference of INO80 to replace H2A.Z with H2A. Our data suggest that INO80 has a mechanism for dimer exchange that is distinct from other chromatin remodellers including its paralogue SWR1.

[1] Department of Epigenetics and Molecular Carcinogenesis, The University of Texas MD Anderson Cancer Center, Smithville, Texas 78957, USA. [2] Center for Cancer Epigenetics, The University of Texas MD Anderson Cancer Center, Houston, Texas 77030, USA. [3] Graduate Program in Molecular Biology, Microbiology and Biochemistry, Southern Illinois University School of Medicine, Carbondale, Illinois 62901, USA. [4] Department of Chemistry and the Huck Institutes of the Life Sciences, The Pennsylvania State University, University Park, Pennsylvania 16802, USA. * These authors contributed equally to this work. † Present addresses: Department of Biochemistry and Molecular Biology, Monash University, Clayton, Victoria 3800, Australia (M.I.U.); Yonsei-IBS Institute, Yonsei University, Seoul 03722, Republic of Korea (J.K.); Division of Hematology, Children's Hospital of Philadelphia, Philadelphia, Pennsylvania 19104, USA and Perelman School of Medicine, University of Pennsylvania, Philadelphia, Pennsylvania 19104, USA (S.K.B.). Correspondence and requests for materials should be addressed to T.-H.L. (email: txl18@psu.edu) or to B.B. (email: BBartholomew@mdanderson.org).

In addition to the primary forms of the four core histones H2A, H2B, H3 and H4 used to package deoxyribonucleic acid (DNA) into chromatin, there are minor histone variants that have important regulatory and biological functions. H2A.Z is one of the most studied histone variants, and has about 60% sequence similarity and non-redundant roles with the canonical histone H2A[1]. H2A.Z is constitutively expressed and incorporated into chromatin by a DNA replication-independent exchange of histone H2A–H2B dimers primarily mediated by the adenosine triphosphate (ATP)-dependent chromatin remodeller SWR1 (or SWR-C)[2]. H2A.Z has been implicated in affecting the stability of nucleosomes[3,4], higher-order chromatin organization[5,6] and the recruitment of transcription factors[7]. H2A.Z is highly enriched at the 5′ ends of genes at the $+1$ and $-1$ nucleosome positions, and has been implicated both in transcription activation and repression[8]. The SWR1 family of remodellers regulates the occupancy and dynamics of the histone variant H2A.Z at gene promoter regions, enhancers, telomeres, centromeres, DNA damage sites and boundary elements[8,9].

SWR1 is a member of a family of ATP-dependent chromatin-remodelling complexes that includes INO80, and similar complexes in humans are SRCAP and human-INO80 (refs 10,11). The SWR1 and INO80 family is characterized by having a split-ATPase (adenosine 5′-triphosphatase) domain interrupted by a long insertion that is unique from the ATPase domains of other Snf2-like helicases[12]. Recruitment of the SWR1 complex is dictated by its affinity for long linker DNA typical of nucleosome-free regions (NFR) and for acetylated histones found at promoter regions[13,14]. The exchange of H2A–H2B for H2A.Z–H2B dimers by SWR1 is mediated in part by delivering of the incoming dimer by one of several histone chaperones (Nap1, Chz1 and FACT)[2,15,16]. SWR1 replaces the H2A–H2B dimer in sequential order in vitro, generating first heterotypic nucleosomes comprising one H2A.Z–H2B and one H2A–H2B dimer[15]. In this process the H3–H4 tetramer is retained even after full replacement of both dimers. Localization of H2A.Z is a very dynamic process, which is controlled not only by its incorporation, but also by its removal. H2A.Z is required for rapid activation of transcription and H2A.Z or Htz1 in yeast is preferentially evicted at promoters when genes are activated[17–21].

There is uncertainty and disagreements on the primary players involved in the removal of H2A.Z. Earlier studies had shown that INO80 can catalyse the exchange of H2A.Z–H2B for H2A–H2B in nucleosomes[22,23], but more recent studies have called this into question[24]. INO80 is a versatile remodelling complex involved in diverse functions such as transcription, DNA repair and replication[11,25,26]. INO80 in yeast activates the INO1 and PHO genes[27,28], stabilizes DNA replication forks during replication stress and is involved in S phase checkpoint control[29–31]. In human cells, INO80 was shown to rapidly remove H2A.Z from sites flanking DNA damage and to facilitate in homologous recombination[32]. Both repair pathways of homologous recombination and nonhomologous end joining in yeast recruit INO80 to double-strand breaks and may remove nucleosomes from these sites[33,34]. Removal of H2A.Z from gene bodies by INO80 has also been implicated in the prevention of non-coding transcription[35]. INO80 not only potentially exchanges histone dimers, but also mobilizes and spaces nucleosomes[36]. Other factors, such as the transcription complex, may have a role as important or more than INO80 in facilitating the exchange of H2A.Z at active promoters[37].

We focused on how INO80 remodels nucleosomes and consequently INO80's regulation of nucleosome dynamics to directly address the role of INO80 in chromatin reorganization. Although the mechanisms of nucleosome spacing, nucleosome

assembly and disassembly by ATP-dependent nucleosome remodellers have been extensively studied[38,39], relatively little is known about the role of ATP hydrolysis and DNA translocation in the SWR1/INO80 family of remodellers for exchanging histone dimers. Remodelling complexes of the ISWI and SWI/SNF families are known to engage and translocate along DNA deep inside of nucleosomes near the dyad axis at superhelical location (SHL) $-2$ (refs 39–42). These remodellers are efficient in repositioning nucleosomes for either nucleosome spacing or disassembly[43]. For these remodelling outcomes, DNA translocation inside of nucleosomes is beneficial as it is thought to pump DNA in and around nucleosomes by creating bulges or small distortions that can move around the nucleosome surface[38,39,44–45]. SWR1 however does not appear to move nucleosomes as a part of exchanging nucleosomal H2A–H2B dimers[2,15]. Different parts of the SWR1 complex have been shown to interact with histone H2A.Z–H2B dimers and to facilitate/regulate their exchange[15,46]. The Swc2 subunit of SWR1 has histone chaperone-like activity that stimulates the incorporation of H2A.Z[22,46,47]. Interactions of SWR1 with the H2A histone fold and nucleosomal DNA at SHL $+2$ are crucial for the activation of SWR1 and suggest that SWR1 translocates along nucleosomal DNA much like SWI/SNF and ISWI[48]. It is however not evident yet how translocation of the ATPase domain on nucleosomal DNA leads or contributes to histone dimer exchange.

There are several preliminary indications that although SWR1 and INO80 belong to the same family of chromatin remodellers they may show significant functional differences. Structural studies have indicated that INO80 and SWR1 interact differently with nucleosomes with SWR1 clinging peripherally to the nucleosome and INO80 grasping the nucleosome more like a hand[49]. The in vivo roles of SWR1 and INO80 are distinct both in transcription and DNA repair, with SWR1 being specialized for incorporation of H2A.Z and INO80 having multiple functions[9,49]. A histone chaperone function is integral to the mechanism of H2A.Z incorporation by SWR1; however, so far INO80 has not been found to contain a subunit like Swc2 with histone chaperone-like activity.

To examine these differences, here we investigate the interactions of Saccharomyces cerevisiae INO80 ATPase domain with nucleosome, the effects of DNA translocation by INO80 on histone–DNA interactions, the role of translocation in altering nucleosome dynamics and its potential in leading to nucleosome movement and dimer exchange. We find that INO80 translocates on nucleosomal DNA at the H2A–H2B dimer interface causing DNA to be persistently displaced from the dimer surface. These changes not only cause mobilization of nucleosomes, but also the preferential exchange of H2A.Z–H2B dimers out of nucleosomes and replacement by H2A–H2B.

## Results

**ATPase domain of Ino80 engages DNA near H2A–H2B.** To gain insight into the mechanism of INO80 remodelling, we first investigated the interactions of INO80 with nucleosomes by identifying the sites in nucleosomal and extranucleosomal DNA protected by INO80 binding. End-positioned nucleosomes with 70 base pairs (bp) of extranucleosomal DNA at one entry/exit site (70N0) were bound to INO80 and accessible DNA sites were cleaved with hydroxyl radicals. Seventy bp long extranucleosomal DNA was shown previously to be optimal for nucleosome binding and mobilization by INO80 (ref. 36). Overlays of the protection patterns of nucleosomes alone and bound to INO80 revealed the regions where INO80 stably interacts with DNA. We found that unlike NURF, ISW2, ISW1a,

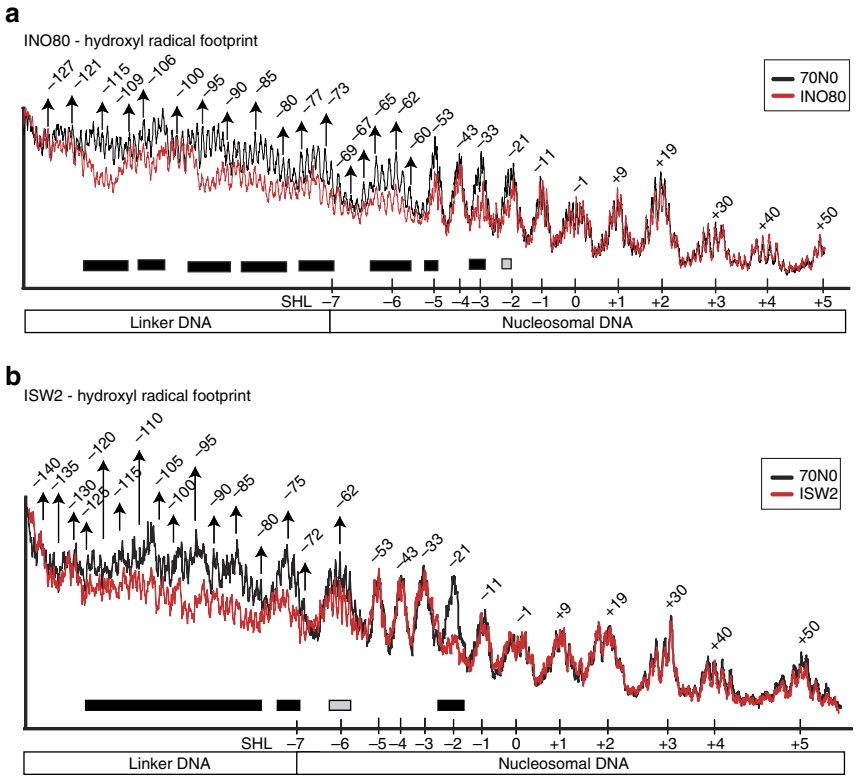

**Figure 1 | INO80 protects nucleosomal DNA at SHL − 3 and SHLs − 5 to − 6.** Hydroxyl radical footprint of (**a**) INO80 and (**b**) ISW2 bound to nucleosomes (red), compared to nucleosome alone. SHL(s) and nucleotides are numbered with reference to the nucleosome dyad axis (0). Regions of strong and intermediate DNA protection are underlined, respectively, with black and grey bars. Data are representative of three technical replicates for both INO80 and ISW2.

SWI/SNF and SWR1; INO80 does not stably bind to SHL − 2, or two DNA helical turns from the dyad axis[39,48]. Instead, INO80 interacts with nucleosomal DNA at SHL − 3, − 5 and − 6 as seen by DNA footprinting protections centred at − 33, − 53 and − 63 nucleotides (nt) from the dyad axis (compare Fig. 1a with Fig. 1b, negative sign refers to the side of the dyad with longer extranucleosomal DNA). INO80 also binds ∼55 bp of extranucleosomal DNA to one side of the DNA helix as shown by the helical periodicity of the DNA protections centred at − 83, − 93, − 103 and − 115 nt from the dyad. Structural modelling of INO80 in complex with nucleosome based on EM, crosslinking and mass spectrometry (CX-MS) suggest that INO80 assumes an elongated conformation, forming a cradle for nucleosome binding[50]. Based on this model, we can envision the Snf2-Ies2, Arp5 and Nhp10 modules to protect nucleosomal DNA at the entry site (SHL7) and the adjoining edge of nucleosomes (SHLs 6, 5), and the Nhp10 module to protect close to SHLs 2, 3 on the other DNA gyre. Similarly, we see strong protection at SHLs − 5 to − 7; however, INO80 does not show stable DNA interactions at SHL 2/3. The protection pattern from hydroxyl radical footprinting suggests that INO80 translocates on nucleosomes at a site other than SHL 2, different from the SWI/SNF and ISWI family of chromatin reemodellers and from even SWR1. Although the contacts for SWR1 (ref. 48) were not examined further as described here for INO80 (see below), DNA gap interference indicates that the ATPase domain of Swr1 is likely bound at SHL + 2 and + 3. The nucleosomal DNA site where the ATPase domain of Ino80 engages is likely near SHL − 5/ − 6 close to the extranucleosomal DNA proximal edge of the nucleosome, or closer to the dyad axis at SHL − 3.

We identified by site-specific DNA crosslinking and label transfer[51], the subunit(s) of INO80 that interact with nucleosomal

DNA at the regions protected from hydroxyl radical cleavage on INO80 binding. A photoreactive deoxyuridine nucleotide analogue with the photoreactive aryl azide tethered optimally 18.6 Å from the nucleotide base (ABG2 probe) was incorporated at target sites in nucleosomal DNA (Fig. 2a compare lanes 2 and 5). We observed the major crosslinking site for Ino80, the catalytic subunit of the INO80 complex, is 58 nt from the dyad (nt − 58), towards the extranucleosomal DNA within the H2A–H2B interface (Fig. 2c lanes 12–14). To determine whether INO80's interactions with nucleosomal DNA varied on ATP binding or ATP hydrolysis, we added either ATP-γ-S (slow hydrolysing form of ATP) or adenosine diphosphate (ADP) to the crosslinking reactions. When a ∼10 Å tether is used, ATP-γ-S and ADP increases the efficiency of Ino80 crosslinking at nt − 58 about four fold, consistent with Ino80 making closer contact with DNA at this position when bound to ATP or ADP (Fig. 2a, compare lanes 2–4, and Fig. 2b). Arp4 is the INO80 subunit that crosslinks to DNA − 30 to − 33 nt from the dyad and not the catalytic subunit as is typical for other remodellers at this nucleosomal region (Fig. 2c, lanes 15–17).

The region of Ino80 crosslinked to DNA at nt − 58 in the presence of ADP is the ATPase domain as confirmed by peptide mapping; an approach similar to that previously used to map interactions of Isw2 and Snf2 with nucleosomes[52,53]. After crosslinking, Ino80 was immobilized onto beads via a C-terminal FLAG tag and partially digested with ArgC protease. C-terminal fragments retained on the beads after ArgC digestion was analysed by SDS–polyacrylamide gel electrophoresis (SDS–PAGE) and immunoblotting using an anti-FLAG antibody. The location of the crosslink to DNA was ascertained by comparing the immunoblot and phosphorimage patterns (Fig. 2d). The results are consistent with Ino80 crosslinking to

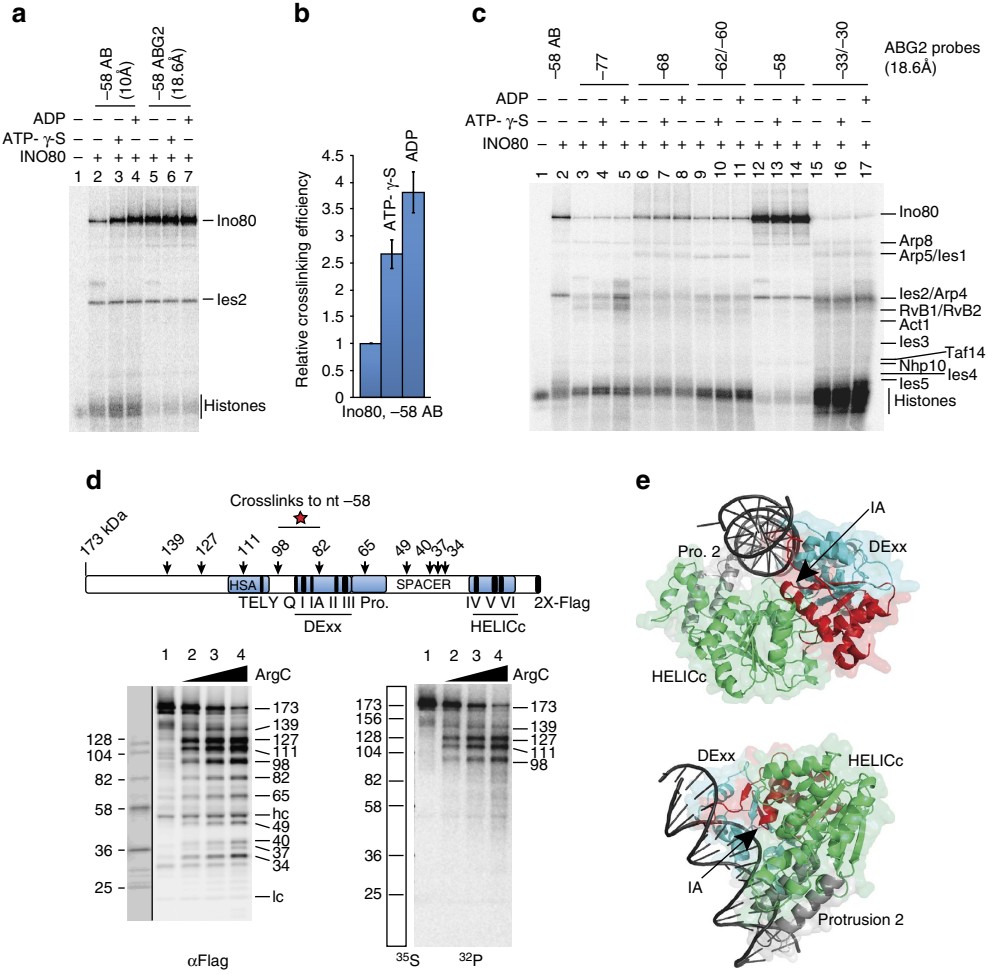

**Figure 2 | The ATPase domain of Ino80 engages with nucleosomal DNA 58 bp from the dyad.** (a–c) Photo-reactive AB-dUMP or ABG2-dUMP nucleotides were incorporated into nucleosomal DNA at nt −58 near SHL −6 (**a**) and other positions (**c**) as indicated for 70N0 nucleosomes. INO80 bound to nucleosomes was crosslinked with or without ATP-γ-S or ADP, as indicated by + or − signs. Ies2 was distinguished from Arp4 by immunoprecipitation of Arp4 from crosslinked samples (Supplementary Fig. 1f) (**b**) The efficiency of the Ino80 catalytic subunit crosslinking to nt −58 with a 10 Å tether (AB-dUMP), −/+ ATP analogues were plotted from panel a, lanes 2–4. Data are representatives of three technical replicates and error bars represent standard deviation of the mean (s.d.) calculated from three technical replicates. (**d**) Peptide mapping of the Ino80 catalytic subunit crosslinked to DNA at nt −58, after digestion of immobilized Ino80 with ArgC. Numbers in the diagram indicate molecular weights of the C-terminal fragments identified by anti-FLAG immunoblotting (αFLAG image). Sites where proteolysis generates these fragments are indicated along with the locations of conserved domains and motifs in Ino80, and the region crosslinked to DNA. In the immunoblot 'hc' and 'lc' indicate, respectively, the antibody heavy and light chains. (**e**) Structural model of the Ino80 ATPase domain bound to DNA. The helicase motif IA is in close proximity with the minor groove of the 3′–5′ DNA strand. The DExx domain is coloured cyan, the HELICc domain green, protrusions 1 and 2 as grey, and the region crosslinked to DNA in red.

DNA in a 145 amino-acid region lying between residues 648 and 793 (corresponding to the 98 and 82 kDa fragments, respectively) (Fig. 2d, compare α-FLAG to [32]P, [35]S ladder, and Supplementary Fig. 1a–d). The same region of Ino80 is crosslinked in the absence of ADP (Supplementary Fig. 1e), demonstrating that this interaction is independent of, but gets stronger on ADP binding. The crosslinked region corresponds to the RecA-like domain I (DExx) of Ino80 encompassing helicase motifs Q, I and IA (Fig. 2d,e), consistent with a homologous region of the ATPase domain of Rad54 from *Sulfolobus solfataricus* found to bind DNA in its crystal structure[54]. Importantly, motif IA of Rad54 has been implicated in DNA binding and generating the power stroke for DNA translocation[54]. Shown previously, SWR1 binds quite differently to nucleosomal DNA than INO80 and is on the opposite side of the dyad at SHL +2 and +3 (ref. 48). CX-MS of INO80 with nucleosomal histones found several interactions of a short conserved region of the Ies2 subunit with the Ino80 ATPase RecA1 and RecA2 lobes[50]. Consistently, we find Ies2 to

specifically crosslink at the same DNA region where Ino80 ATPase is bound, and not in other positions along the nucleosomal DNA (Fig. 2a and Supplementary Fig. 1f).

**Translocation by INO80 near H2A–H2B mobilizes nucleosomes.** All ATP-dependent chromatin-remodelling complexes belonging to the SF2 family of helicases share a common property of DNA translocation[39,44]. Typically translocation of chromatin remodellers can be blocked by as little as a one bp gap in DNA or a nick in DNA can interfere with chromatin remodelling by prematurely releasing DNA twist that is needed to create the torsional force sufficient to move DNA around nucleosomes[40–42]. The site where DNA translocation occurs has been mapped in some instances by generating a pool of DNA in which a one nucleotide gap is present in one strand of each DNA molecule at a different position. The pool of DNA templates is used to select for those DNA templates where the gap is at the right position to

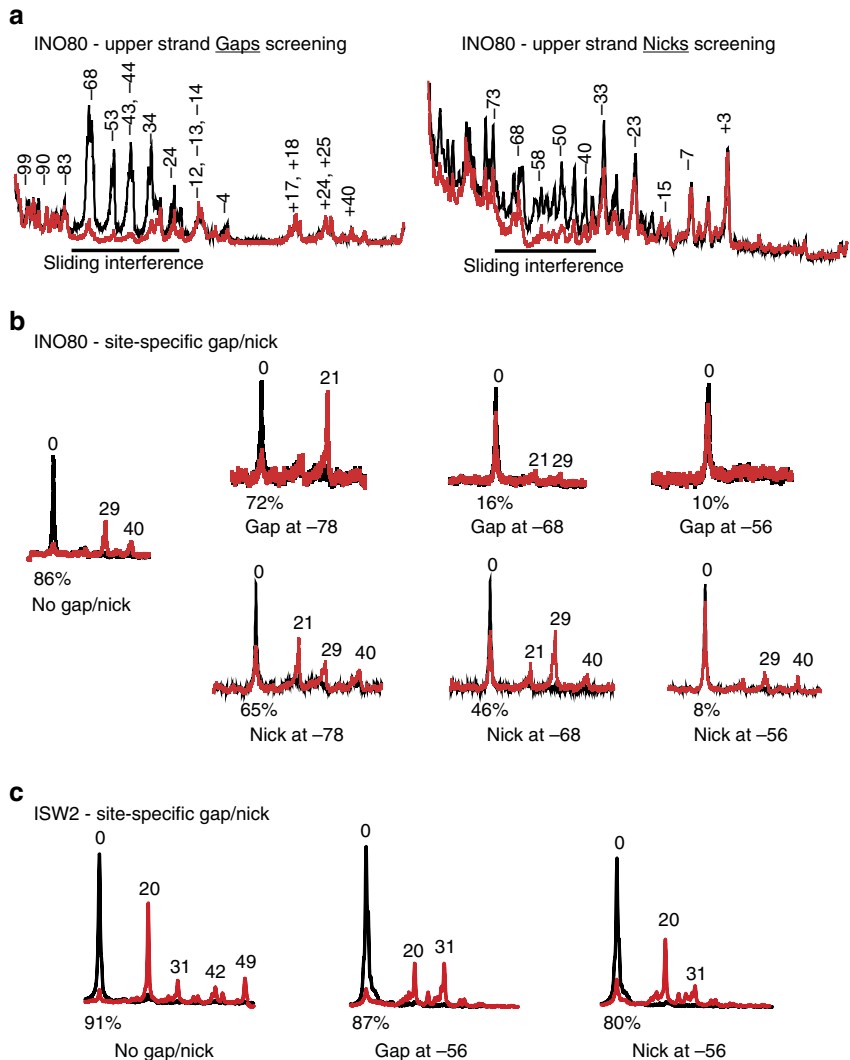

**Figure 3 | Single nucleotide gaps and nicks in DNA at the H2A–H2B interface interfere with nucleosome mobilization by INO80. (a)** The position(s) where DNA gaps and nicks interfere with INO80 remodelling were screened using 0N70 nucleosomes containing random gaps or nicks. The overlays show enrichment of DNA containing gaps or nicks at indicated positions with nucleosomes that were mobilized (red) versus those that were not (black). Nucleotide positions of the gaps and nicks are indicated relative to the dyad axis (0). Gaps and nicks that interfered with remodelling (underlined) were enriched in the immobile fraction. **(b)** Nucleosomes with a gap or nick in DNA at the specified positions in the upper strand were remodelled with INO80. Changes in nucleosome position were mapped by site-directed cleavage. The cleavage sites were mapped before (black) and after (red) INO80 remodelling[57]. The original position of the nucleosome is denoted as 0 and numbers indicate bp moved. Fractions (%) of the original nucleosome position lost due to remodelling are indicated. **(c)** Effects of a single gap or nick at nt − 56 on ISW2 remodelling were examined as for INO80 (refer to **b**). Data are representative of four technical replicates for **a** and three technical replicates for **b,c**.

block translocation versus all other positions where it will not. If DNA translocation by INO80 occurs near SHL − 5/− 6 where the ATPase domain engages, we expected that DNA gaps at this region should interfere with remodelling[40,42,55,56]. This approach showed that gaps in the upper strand from − 34 to − 68 nt and to a lesser extent in the lower strand from − 27 to − 64 nt from the dyad block INO80 from mobilizing nucleosomes (Fig. 3a and Supplementary Fig. 2a). These patterns suggest that INO80 has a mild strand preference for translocation and that it is unlike other remodellers where gaps 15–35 nt from the dyad interfered[40–42,48]. The gap interference pattern is consistent with the ATPase domain of INO80 binding to DNA at the H2A–H2B interface and contrasts with the ATPase domains of SWI/SNF and ISW2 binding to DNA at the H3–H4 interface. These data suggest the specificity of INO80 for histone dimer exchange may be related to its position of translocation on DNA.

Another approach for mapping where the chromatin remodeller translocates on DNA is to place a one nucleotide gap in DNA at a position distal from the active site of the enzyme and track DNA movement as remodelling occurs[57]. As remodelling proceeds, DNA movement will be arrested when the gap reaches the active site and DNA movement can be followed with bp resolution. When no gap is present INO80 moves DNA 29 and 40 nt from its starting position at the DNA entry site (upper strand) into nucleosomes or 36 nt out of the nucleosomes at the DNA exit site (lower strand) as shown by crosslinking and subsequent cleavage (Fig. 3b and Supplementary Fig. 2b). In contrast, DNA is moved at the entry site only 21 nt when a gap is at nt − 78, thus stopping when the gap reaches where the Ino80 ATPase domain is bound, 57 nt from the dyad (Fig. 3b). DNA gaps near positions where the ATPase domain binds, at nt − 46, − 56 and − 68 block INO80 completely from moving

DNA inside of nucleosomes, consistent with our previous scan (Fig. 3b and Supplementary Fig. 2b,c). We compared the gap interference of INO80 with that of ISW2. For ISW2, it is known that DNA translocation for nucleosome mobilization starts near nt $-17/-18$ close to SHL $-2$ (refs 42,52). A DNA gap at nt $-56$ should not interfere with nucleosome mobilization by ISW2 as it does for INO80. When no gap is present, ISW2 moves nucleosomes predominantly 20 nt and to lesser extent up to 49 nt into nucleosomes at the DNA entry site. Unlike INO80, ISW2 continues to move nucleosomes with a gap at nt $-56$ until the gap is 25 or 36 nt from the dyad, consistent with the gap reaching where the ATPase domain is bound at SHL $-2$ (Fig. 3c). Therefore, in contrast to other remodellers like ISW2, DNA translocation of INO80 at the H2A–H2B interface is required for INO80-mediated nucleosome movement.

Next, we investigated the role of DNA twist by releasing torsional strain with nicks in DNA. DNA nicks can block nucleosome movement by inhibiting the accumulation of DNA twist and unlike gaps do not typically block translocation[58–60]. A scan for nicks either in the upper or lower strands of DNA show that nicks $-40$ to $-63$ nt from the dyad block INO80-mediated nucleosome movement comparable to that observed with gaps (Fig. 3a and Supplementary Fig. 2a). Nicks however if moved into this region during remodelling have only marginal effects. Close to the DNA entry site, a single nick at $-68$ or $-78$ nt from the dyad in the upper strand weakly alters nucleosome movement, but still moves 29 and 40 nt from the dyad (Fig. 3b). At nt $-68$ and $-78$, there is a clear difference in the effect of a DNA gap (extensive blocking at distinct steps) versus a nick (minimal blocking). Nicks from nt $-36$ to $-56$ dramatically block nucleosome movement, comparable to a gap, and nicks at nt $-16$ and $-25$ do not, a property not shared with ISW2 (Fig. 3b,c and Supplementary Fig. 2b,d). The nick interference pattern indicates that DNA twists and torsional strain are required in the region spanning the H2A–H2B interface to initiate, but not afterwards in continued nucleosome movement. This suggests that initiating nucleosome movement requires torsional strain to potentially displace DNA from the H2A–H2B interface, which can presumably be maintained more readily afterwards. A nick in DNA at nt $-56$ affected ISW2-mediated nucleosome movement the same way as with a single nucleotide gap, showing a clear difference from INO80 in its mechanism of nucleosome mobilization (Fig. 3c). Our data shows that INO80 has a unique way to mobilize nucleosomes distinct from other known ATP-dependent chromatin remodeller and may represent a unique way to exchange histone dimers that is distinct from SWR1.

**INO80 persistently displaces DNA from the H2A–H2B interface.** We envisioned that INO80-mediated nucleosome movement or dimer replacement could involve distortions of DNA or DNA bulges at the dimer surface. Persistent distortions at the H2A–H2B interface would increase the accessibility to and the potential for displacing H2A or H2A.Z dimers from nucleosomes. We followed DNA movements by means of mapping the contacts between p-azido phenacyl bromide (APB)-conjugated amino-acid residue 53 of histone H2B and DNA at the two edges of the nucleosome[57]. Modified H2B53 crosslinks to one strand of DNA 54 nt from the dyad at the H2A–H2B interfaces on either half of the nucleosome. Snapshots of DNA movements near where INO80 translocates on DNA and DNA enters nucleosomes revealed that DNA is initially moved 11 and 20 nt with 20 nt being the predominant step, and later to a lesser extent, 26 and 30 nt (Fig. 4a, lanes 1–8, and Supplementary Fig. 3a). We observed an overall loss of DNA contacts with H2B as seen by the

loss of signal at the starting position relative to the gain of signal at the new sites (Fig. 4b). The signal loss attributed to a loss of DNA contacts with histone H2B is specific to INO80, because there is no progressive loss of DNA contacts when ISW2 mobilizes nucleosomes (Fig. 4c). The progressive reduction in H2B contacts is also specific to the side of nucleosomes where INO80 translocates on nucleosomal DNA, and is not observed on the other side of nucleosomes where DNA exits during remodelling, showing that large DNA distortions are localized at the region of translocation (Supplementary Fig. 3c). The larger steps of DNA movement observed with INO80 compared to ISW2 are consistent with INO80 disrupting H2B contacts with DNA, combined with the latter rebinding of DNA to the histone octamer. ISW2 step sizes of a $\sim 7$ nt initial step followed by multiple $\sim 2$ nt steps (Fig. 4a, lanes 9–16) shows that ISW2 generally moves DNA inside of nucleosomes at this position without making distortions larger than $\sim 2$ bp, consistent with previous data[61]. Our results suggest that INO80-mediated replacement of H2A.Z in nucleosomes would preferentially occur on the side of nucleosomes where there is sufficient extranucleosomal DNA (Fig. 7). Given INO80 *in vivo* binds to the NFR and $+1$ nucleosomes, our data are consistent with the *in vivo* pattern of H2A.Z in which the NFR proximal side of the $+1$ nucleosome is preferentially H2A.Z deficient compared to the NFR distal side[62].

**INO80 requires DNA translocation to replace H2A for H2A.Z.** Given the persistent disruption of histone–DNA contacts at the H2A–H2B interface, we examined by single-molecule Förster resonance energy transfer (smFRET) the potential of INO80 mediating dimer exchange and it's selectivity for H2A.Z versus H2A. Three-colour smFRET is used to monitor exchange between H2A.Z–H2B and H2A–H2B, with H2B labelled with different acceptors (ATT 0647N and Cy5.5) depending on whether assembled with H2A.Z or H2A and a FRET donor (Cy3) attached to DNA (Fig. 5a and Supplementary Figs 4 and 5). We observed that H2A.Z–H2B dimers are exchanged out of nucleosomes with INO80 and replaced with H2A–H2B dimers only in the presence of ATP, and that the converse exchange of H2A for H2A.Z is negligible. In the presence of ATP, H2A is $\sim$ nine times more likely to be incorporated in place of H2A.Z than the converse (10.3% versus 1.2%) or in the absence of ATP (1.6%) (Fig. 5b,c). To prove that DNA translocation and twist at the H2A.Z–H2B dimer interface are both required for exchange mediated by INO80, we introduced a gap or a nick, respectively, at nt $-56$. We found that either perturbation decreased the dimer exchange efficiency close to no ATP level, indicating that nucleosomal DNA translocation by INO80 near SHL $-6$ is coupled to its dimer exchange activity (Fig. 5c). Our results validate that INO80 specifically exchanges H2A.Z–H2B dimers for H2A–H2B in an ATP-dependent manner, and is dependent on nucleosomal DNA translocation and the accumulation of DNA torsional strain at the region of translocation.

**INO80 preferentially mobilizes H2A.Z nucleosomes over H2A.** The H2A.Z selectivity of INO80 could in part be due to enhanced translocation and DNA dissociation at the H2A–H2B dimer when nucleosomes contain H2A.Z instead of H2A. To test this, we compared INO80-mediated mobilization of H2A.Z versus H2A nucleosomes, and the affinity of INO80 for the two kinds of nucleosomes. Consistent with our hypothesis, we found that INO80 moves nucleosomes with H2A.Z about four times faster than with H2A (Fig. 6a and Supplementary Fig. 6a,b). INO80 bound to H2A.Z nucleosomes with an affinity comparable to H2A nucleosomes as seen by electrophoretic-mobility shift in

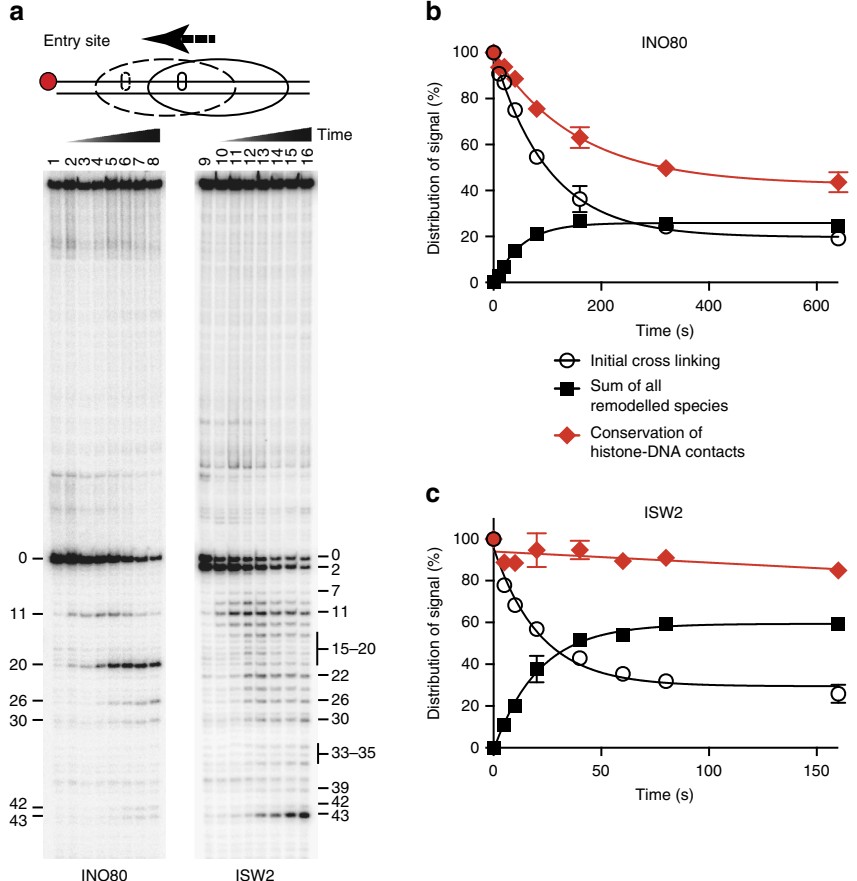

**Figure 4 | Translocation of INO80 persistently displaces DNA from the H2A–H2B interface.** (**a**) Nucleosomes modified at amino-acid residue 53 of H2B were used to examine movements of DNA on the octamer surface at the edges of nucleosomes. Shown here are DNA movements at the longer extranucleosomal DNA proximal edge (entry site) of nucleosomes. DNA in nucleosomes without remodelling are cleaved 54 bps from the dyad axis and are labelled as 0. Shorter DNA bands appear due to DNA moving into nucleosomes and across residue 53 of histone H2B. Numbers beside the bands correspond to the number of nucleotides moved from the starting position. (**b,c**) The amount of DNA cleaved at the starting position (initial crosslinking) and the total amount of those bands that accumulate during remodelling (sum of all remodelled species) were plotted versus time for INO80 (**b**) and ISW2 (**c**). Error bars represent s.e.m. calculated from three technical replicates.

native gel (Supplementary Fig. 6c,e). The affinity of INO80 for H2A.Z and H2A nucleosomes were further examined by competitive binding with free DNA, which is a more stringent assay to measure relative differences in affinity[63]. INO80 even under these conditions did not show any binding preferences for H2A.Z over H2A nucleosomes (Supplementary Fig. 6d,f). The increase in nucleosome movement with H2A.Z is also not due to a higher rate of ATP hydrolysis because INO80 only show marginal difference in the rates of nucleosome-stimulated ATP hydrolysis with H2A.Z versus H2A nucleosomes (Supplementary Fig. 6g). The higher rate of H2A.Z nucleosome mobilization is unlikely to be due to an intrinsic increase in nucleosome mobility when H2A.Z is present[64], because no differences were observed with ISW2 moving H2A.Z versus H2A nucleosomes (Fig. 6b). A more likely reason for the specificity of INO80 for H2A.Z nucleosomes could be that there is a difference in the interactions of INO80 with H2A.Z versus H2A nucleosomes. However, no H2A.Z-specific-binding motif in INO80 is currently known, unlike the α2-helix of the canonical histone H2A that has been shown to activate the exchange of H2A–H2B dimers for H2A.Z–H2B by SWR1 (ref. 48).

**INO80 moves nucleosomes less efficiently than ISW2.** We compared the efficiencies of nucleosome movement by INO80 and ISW2 to find out if differences in the sites of translocation on nucleosomal DNA affect the efficiency of nucleosome movement. Both INO80 and ISW2 move nucleosomes towards the centre of DNA making it easier to compare. INO80 repositioned nucleosomes ∼80 times slower than ISW2 even though INO80 hydrolysed ATP only two times slower under these conditions (Fig. 6c–e and Supplementary Fig. 6h). These data suggest that INO80 is dramatically less efficient in converting ATP hydrolysis into nucleosome movement than ISW2. One reason INO80 may be less efficient at mediating nucleosome movement is that distortions at the H2A–H2B interface may not as readily propagate through nucleosomes, and therefore persist longer than distortions at the H3–H4 interface (Fig. 7). Therefore, DNA translocation at the H2A–H2B interface is likely less optimal for repositioning or sliding nucleosomes than translocation at the H3–H4 interface.

## Discussion
The INO80 family of ATP-dependent chromatin remodellers consisting of INO80 and SWR1 in yeast is distinctive among the wide variety of remodelling complexes. They are the only class of remodellers known to catalyse the ATP-dependent exchange of H2A–H2B dimers without disrupting the entire nucleosome and enrich nucleosomes for a particular H2A variant[2,23,46,65].

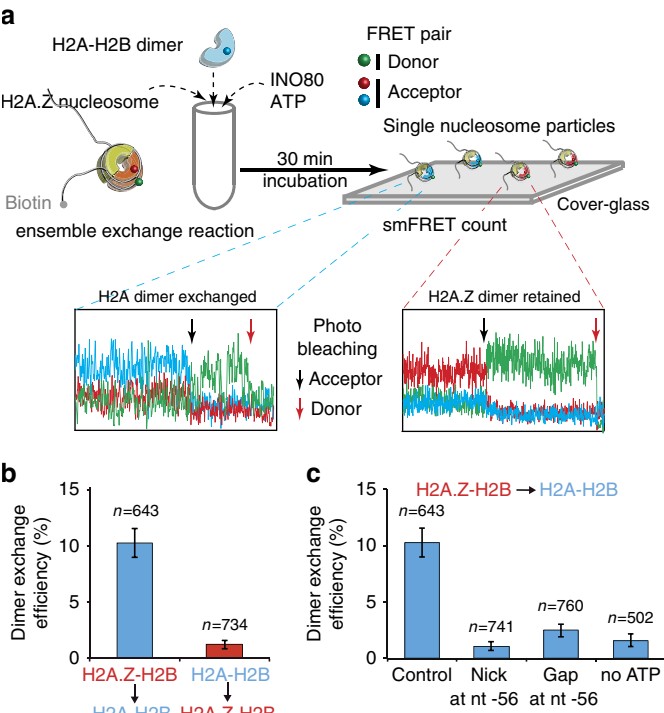

**Figure 5 | INO80 requires DNA translocation for exchanging H2A.Z for H2A in nucleosomes.** (**a**) Experimental scheme to determine the efficiency of INO80 for dimer exchange at single-nucleosome level, based on three-colour FRET. Nucleosomes forming FRET pairs with Cy3 (DNA) and Cy5.5 (H2A–H2B dimer) or ATTO 647N (H2A.Z–H2B dimer) were counted to measure the efficiency of dimer exchange. The acceptor is photo-bleached at a single time point (black arrow), which is followed by a single-step donor (Cy3) photobleaching (red arrow). (**b,c**) The results are shown for efficiencies of H2A–H2B incorporation and H2A.Z–H2B incorporation (**b**), and the effects of a single nucleotide gap or nick at nt − 58 or the absence of ATP on H2A–H2B incorporation (**c**). The number of single molecules (*n*) analysed for each reaction condition is shown above its corresponding bar.

We have found that for INO80 these differences are evident in its mechanism of nucleosome mobilization. Several groups, including ours, have shown before that RSC, SWI/SNF, ISW2 and NURF mobilize nucleosomes by translocating on DNA at the SHL 2 position inside of nucleosomes near the dyad axis[40–42]. Although SWR1 is not able to reposition nucleosomes like the remodellers mentioned above, a recent study indicates that SWR1 also might translocate on DNA close to the centre of nucleosomes[48]. We have now found that not only is the mechanism of INO80 for exchanging histone dimers distinct from that of chromatin remodellers whose primary role is to mobilize, space and disassemble nucleosomes (that is, ISWI and SWI/SNF)[39], but is also different from its paralogue SWR1 (ref. 48). This study demonstrates that unlike any other remodelling complex known to date, INO80 translocates on DNA close to the extranucleosomal DNA proximal edge of nucleosomes at the H2A–H2B interface for nucleosome mobilization, and in the process disrupts histone–DNA interactions at the H2A–H2B interface for extended times. INO80 utilizes a direct approach to promote dimer exchange, which entails persistently disrupting histone–DNA interactions at the H2A–H2B for dissociation of these dimers without displacing the entire nucleosome (Fig. 7). Nucleosome interactions and translocation of the INO80 ATPase makes this remodelling complex uniquely adapted for changing nucleosome dynamics in terms of the H2A/Z–H2B dimers.

Our data are supported by the CX-MS experiments of the Hopfner group[50] showing the ATPase domain of Ino80 close to the H2A–H2B dimer and the DNA entry site of nucleosomes. They showed the RecA2 lobe of Ino80 crosslinks to the L2 loop region of histone H2A, adjacent to SHL − 6, and the insertion within RecA2 to the N terminus of H3 near SHL − 6.5. The cryoEM data suggest INO80 has a hinge region where the ATPase domain resides and the hinge connects two large lobes that cradle nucleosomes and places the SHL 5 and 6 positions close to the hinge region. This model of INO80 interactions with nucleosomes coincide remarkably well with where we find the ATPase domain of Ino80 bound to nucleosomal DNA. We find that INO80 utilizes DNA twist and torsional strain for moving DNA around nucleosome and for H2A.Z–H2B dimer exchange. Generation of torsion in DNA by positive torque (supercoiling) has been reported to specifically induce H2A/H2B dimer loss without disrupting $(H3/H4)_2$ tetramer–DNA interactions or causing unwrapping of nucleosomes[66]. The H2A.Z versus H2A specificity of INO80 for dimer exchange is in part conferred by how efficiently DNA translocation and the displacement of DNA from the dimer surface can occur in nucleosomes containing H2A.Z-H2B or H2A-H2B dimers. SWR1 specificity on the other hand is tied to free H2A.Z–H2B and not H2A–H2B dimers stimulating SWR1 activity along with the chaperone Swc2 helping to lock-in the incoming dimer[48]. These intrinsic differences in their mechanisms may account for SWR1 being more highly specialized than INO80 and for them having distinctly different effects on DNA repair[49]. The precision of the dimer exchange mechanism of INO80 may make the removal of H2A.Z by INO80 less robust than the elongating RNA polymerase II complex that can completely remove H2A.Z nucleosomes including the H3–H4 tetramer[37]. While INO80 may not be the primary complex to remove H2A.Z at promoters[37], it may be a major factor for removing H2A.Z at DNA damage sites or at centromeres and telomeres. Our observation of the less efficient mobilization of nucleosomes by INO80 compared to ISW2 is also consistent with *in vivo* mapping of nucleosome positioning. Reduction of INO80 does not have as significant effect on nucleosome positioning as does loss of ISW2 or ISW1 and yet causes nucleosomes to be more 'fuzzy'[37], consistent with the lowered efficiency of INO80 to mobilize nucleosomes.

Most previous studies have used epitope or fluorescent tags on histones in ensemble experiments to track the exchange of H2A and H2A.Z using gel shift-based assays[15,22–24]. These assays appear to provide varied results that differ from lab to lab. The variations might be due to the ensemble-averaging nature of the assays that cannot accurately normalize the results from a heterogeneous mixture of nucleosomes, sub-nucleosomes and aggregates. The smFRET approach used here enables monitoring of individual nucleosomes, thereby providing a robust and accurate assay for the exchange of histone dimers. Much like the nucleosome mobilization efficiency differences between INO80 and ISW2, the efficiency of dimer exchange however may be different between INO80 and SWR1.

ATP-dependent chromatin remodellers exploit the biophysical properties of regions at SHL 2 and 5 in nucleosomes that can accommodate distortions such as over- or under-winding of DNA as has been observed in their crystal structures[67,68]. Previously, all remodellers that had been studied were found to take advantage of this flexibility at SHL − 2, and now we have found a remodeller that utilizes the DNA torsional flexibility at SHL − 5 to mobilize nucleosomes and exchange histone dimers. This is also a unique ATP-dependent chromatin remodeller, which invades nucleosomes at the entry site as originally proposed by Peter Becker and colleagues nearly a decade and a half ago[69] (Fig. 7).

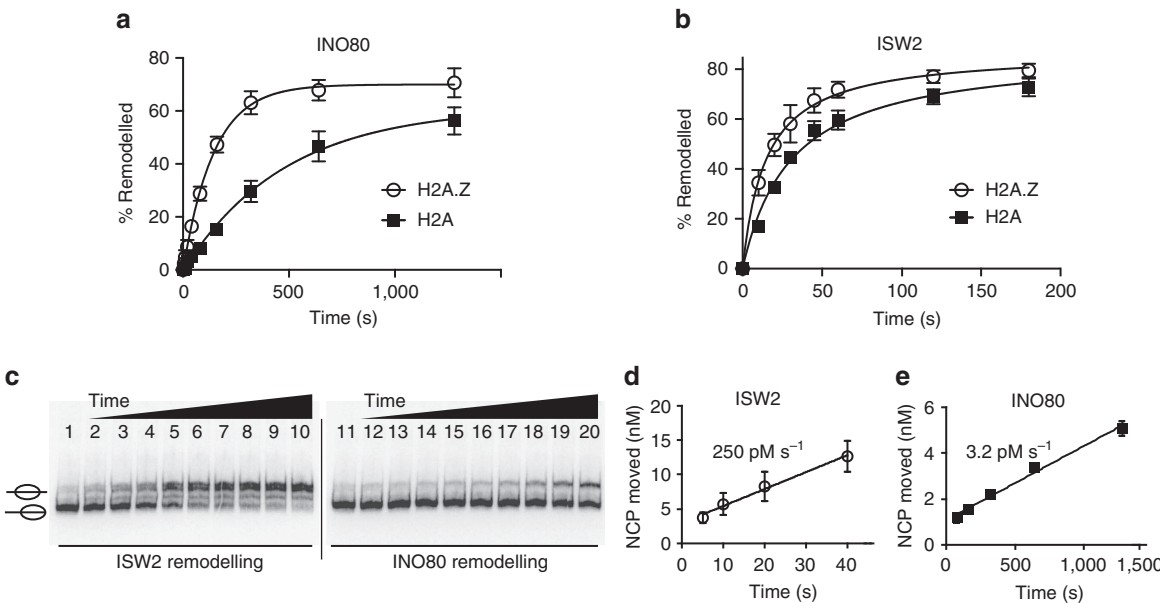

**Figure 6 | INO80 preferentially mobilizes H2A.Z over H2A nucleosomes.** (**a,b**) The rates of nucleosome movement by (**a**) INO80 and (**b**) ISW2 with H2A.Z and H2A nucleosomes as determined by electrophoretic-mobility shift assay in native gels (Supplementary Fig. 6a). The fractions (%) of nucleosomes moved were plotted versus time to determine the rates of H2A.Z versus H2A nucleosome movement. (**c**) The efficiencies of ISW2 (lanes 2–10) and INO80 (lanes 11–20) for mobilizing nucleosomes were determined by the rates with which they repositioned H2A-containing nucleosomes. End-positioned (70N0) nucleosomes were completely bound with ISW2 or INO80 and remodelled with 80 µM ATP at 18 °C for 0, 5, 10, 20, 40, 80, 160, 320, 640 and 1,280s. (**d,e**) The amounts of nucleosomes moved were plotted versus time to determine the initial rates of nucleosome movement by ISW2 (**d**) and INO80 (**e**). NCP is for nucleosome core particles. Error bars represent standard deviation of the mean (s.d.) calculated from three technical replicates.

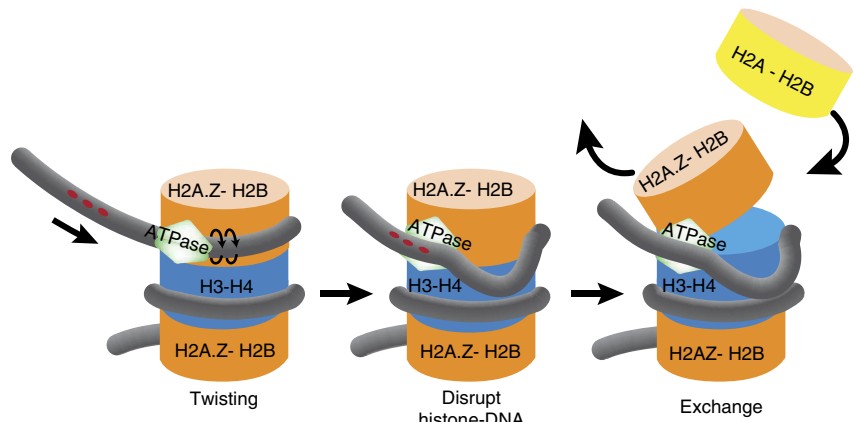

**Figure 7 | INO80 translocates along DNA at the H2A–H2B dimer interface to displace DNA and promote H2A.Z exchange.** A model showing DNA translocation by INO80 close to the DNA entry site of nucleosomes for nucleosome mobilization and histone dimer exchange. Translocation persistently destabilizes dimer–DNA interactions and promotes the exchange of H2A.Z–H2B dimer.

## Methods

**Enzyme purification.** INO80 and ISW2 complexes were purified from *Saccharomyces cerevisiae* strains S288C[70] and YTT480 (ref. 71), respectively. Briefly, double FLAG epitope tags were inserted at the 3′-end of the *INO80* and *ISW2* genes and the complexes were purified by immunoaffinity chromatography using ANTI-FLAG M2 Affinity Gel (Sigma) and eluted with FLAG peptides. Complex purity and integrity was determined by analysing samples on 4–20% SDS–polyacrylamide gels. Enzyme complexes were tested for nucleosome binding, remodelling and nucleosome-stimulated ATP hydrolysis activity[36].

**Nucleosome reconstitution.** Mono-nucleosomes ('70N0' or '0N70', N = nucleosome core particle with 147 bp of 601 DNA and number indicated length of extranucleosomal DNA) were assembled at 37 °C by salt dilution with 7–10 µg of recombinant *Xenopus laevis* octamers (wild type or cysteine mutant octamer), 100–200 fmol of 5′ $^{32}$P-labelled 601 nucleosome-positioning sequence[57,72] containing DNA probes, 5–10 µg of sheared salmon sperm DNA (heterogeneous assembly) or polymerase chain reaction (PCR) amplified 601 DNA

carrier (homogeneous assembly, for ATPase assay only)[36]. DNA and histone octamer were mixed at roughly 1:1 w/w ratio in 2M NaCl, and the salt concentration was diluted in steps down to 280 mM with 25 mM Tris-HCl (pH 8). Samples were run on 4% native polyacrylamide gel to check for nucleosome assembly.

**Hydroxyl radical footprinting.** DNA with a 5′ $^{32}$P-label on one strand was assembled into end-positioned nucleosome with 70 bp of extranucleosomal DNA (70N0). INO80 and ISW2 were bound to nucleosomes in saturating amounts in low-glycerol conditions (<0.8%) at 30 °C. Cleavage reactions were initiated by the addition of 280 µM Fe(II), 0.17% H$_2$O$_2$, 5.7 mM ascorbate and 220 µM EDTA; and were incubated for 30 s. Reactions were terminated with the addition of 100 µl of 5 M ammonium acetate, 5 mM thiourea and 10 mM EDTA. The stability of the bound complex was checked before and after cleavage by native gel shift assays. DNA was isolated by phenol-chloroform extraction, concentrated by ethanol precipitation at − 20 °C, and resolved in denaturing 6.5% polyacrylamide gels with 8 M urea along with a sequencing ladder made from the same DNA. The data

were analysed with ImageQuant. Microsoft Excel was used to plot overlays of nucleosome alone and INO80/ISW2-bound signals after normalization of signals based on total DNA.

**Site-specific DNA photoaffinity crosslinking.** Site-specific photoreactive DNA probes were synthesized by enzymatic incorporation of modified nucleotides into double-stranded DNA[51]. dUMP analogues coupled with *p*-azidophenacyl bromide with a chain length of 10 Å (AB probes) or 18.6 Å (ABG2 probes) were incorporated along with [$\alpha$-$^{32}$P] dGTP/dATP in tandem at specific positions. Photoreactive DNA was reconstituted into 70N0 nucleosomes and bound with saturating amounts of INO80 at 30 °C. The extent of INO80 binding was assessed on 4% native polyacrylamide gels. In all experiments, >90% of nucleosomes were bound by INO80. After binding, INO80 was crosslinked to DNA by ultraviolet irradiation (3 min at 310 nm, 2.65 mW cm$^{-2}$), and DNA was digested with DNase I and S1 nuclease for transfer of the radioactive label to the crosslinked protein(s)[51]. Protein subunits were separated on 4–20% SDS–polyacrylamide gels and radiolabelled subunit(s) were visualized by phosphorimaging. Data are representatives of three technical replicates.

**Peptide mapping of crosslinked Ino80 subunit.** Photoaffinity-labelled INO80 complex (after digestion of DNA and label transfer) was denatured with 0.4% SDS and heating at 90 °C for 3 min, followed by buffer exchange using Amicon Ultra filters to remove SDS and FLAG peptides. C-terminal FLAG-tagged Ino80 was purified by immobilization on ANTI-FLAG M2 Affinity Gel (Sigma). Protein-bound beads were washed and resuspended in ArgC incubation buffer containing 50 mM Tris-HCl (pH 7.8), 5 mM CaCl$_2$ and 2 mM EDTA. Protein cleavage was initiated by the addition 5 mM DTT (final concentration) and varying concentrations of ArgC protease (Promega, sequencing grade) with incubation at 37 °C for 2 h. Reactions were stopped by the addition of 1 mM PMSF and 10 mM EDTA. Immobilized C-terminal fragments were separated from the released fragments and washed three times in the same buffer as the digestion. The bead fractions were resuspended in SDS–PAGE sample buffer and analysed on 4–20% Tris-glycine SDS–polyacrylamide gels by anti-FLAG immunoblotting using mouse monoclonal ANTI-FLAG M2-Peroxidase (HRP) antibody (Sigma-Aldrich catalogue # A8592) in 1:1,000 dilution, and phosphorimaging. Apparent molecular masses of the Ino80 fragments were estimated by comparing their migration relative to the $^{35}$S-labelled Ino80-FLAG markers of known molecular weights prepared by *in vitro*-coupled transcription and translation using TnT T7 Quick Coupled Transcription/Translation System (Promega) (Fig. 2d, lines alongside the images for $\alpha$FLAG and $^{32}$P represent migration of the [$^{35}$S]-methionine labelled peptide markers on SDS–PAGE and their molecular weights).

**Western blot analysis.** Protease (ArgC) digested Ino80 fragments were resolved in 20 cm X 20 cm 4–20% Tris-glycine SDS–polyacrylamide gels and transferred onto PVDF membranes using Bio-Rad Trans-Blot electrophoretic transfer cell for 3 h at 4 °C, using a transfer buffer containing 25 mM Tris, 192 mM glycine, 20% methanol, 0.1% SDS (pH 8.3) and 50 V (constant). The membranes were blocked with 5% fat-free milk in TBST (20 mM Tris-HCl pH 7.5, 150 mM NaCl, 0.1% Tween-20), overnight at 4 °C, washed with TBST, and incubated with mouse monoclonal ANTI-FLAG M2-Peroxidase (HRP) antibody (Sigma-Aldrich catalogue # A8592) in 1:1,000 dilution for 1 h at room temperature. The blots were washed with TBST, and developed with SuperSignal West Femto Maximum Sensitivity Substrate (Thermo Fisher) with Image Quant LAS 4000 (GE Healthcare Life Sciences) (Supplementary Fig. 7).

**Synthesis of Ino80-FLAG peptide markers.** The *INO80* gene with a FLAG tag at the 3′end, along with 100 bps of DNA flanking each end of the coding sequence was PCR amplified from genomic DNA isolated from the same yeast strain used for INO80-FLAG purification. This DNA was used as the template for PCR amplification of gene constructs with segments of *INO80-FLAG* of varying lengths from the 3′end. Ino80-FLAG peptides of varying lengths from the C terminus were synthesized using the TnT Quick Coupled Transcription/Translation System (Promega) with [$^{35}$S]-methionine (Perkin Elmer). Transcription/translation reactions were carried out at 30 °C for 90 min, followed by the addition of 20 μg ml$^{-1}$ leupeptin and analysed on 4–20% Tris-glycine gradient SDS–polyacrylamide gels and phosphorimaging. Peptide markers of all sizes were pooled and run alongside Ino80 peptide mapping reactions. Relative electrophoretic mobilities ($R_f$) of the peptide marker from three technical replicates were calculated and plotted against log$_{10}$ of their molecular weights to generate a standard curve (Supplementary Fig. 1a,b). Molecular weights and sizes of protein fragments generated by ArgC cleavage were estimated from the standard curve based on their $R_f$. Protease cleavage sites were mapped to arginine residues closest to the estimated cleavage sites.

**Modelling of Ino80 ATPase domain.** A predicted structure of the Ino80 ATPase domains was obtained using Phyre$^2$ protein homology modelling tool based on hidden Markov model[73]. The structure of the *Sulfolobus solfataricus* Rad54 ATPase

domain in complex with 25-bp double-stranded DNA (PDB ID 1Z63)[54] was used as the structural template for modelling the Ino80 ATPase associated with DNA.

**Synthesis of DNA probes with gaps and nicks.** DNA probes (217 bp) containing the Widom 601 nucleosome-positioning sequence (147 bp)[73] followed by 70 bp of extranucleosomal DNA (0N70), with a 5′-$^{32}$P label on the upper or lower strand was synthesized separately by PCR. For incorporating random single nucleotide gaps in DNA, PCR reactions were supplemented with 2–20 μM dUTP. Uracil was removed from DNA by uracil-DNA glycosylase (NEB), followed by nicking DNA at the abasic sites with endonuclease III (ref. 42). Cleaved DNA were resolved in denaturing 6.5% polyacrylamide gels with 8 M urea along with ddT sequencing ladder to confirm the positions of the incorporated uracil residues and single nucleotide gaps. Random single nicks on the DNA backbone were generated by digestion with an appropriate amount of DNase I to create single cuts[42].

DNA probes with site-specific gaps and nicks were made by immobilization of 3′-biotinylated DNA template strand onto streptavidin-Dyna beads M280 (Dynal Biotech). The template strand was 5′-$^{32}$P-labelled. Two fragments of DNA flanking the desired gap or nick were synthesized by PCR. One of the PCR primers for generating the flanking fragments of DNA was phosphorylated. Both DNA fragments were digested with lambda exonuclease (NEB), which preferentially cleaves the phosphorylated strand. Single-stranded DNA fragments flanking the gap or nick at specific sites were annealed to the template by incubating at 95 °C with slow cooling to 37 °C. DNA was released from the beads by cutting with the appropriate restriction endonuclease.

**DNA gaps and nicks interference with nucleosome mobilization.** Nucleosomes (0N70) with gaps or nicks in DNA were remodelled with INO80 to achieve movement of ~50% of the nucleosomes. Remodelled (mobile) and un-remodelled (immobile) nucleosomes were resolved on preparative 5% native polyacrylamide gels and purified by passive elution, followed by phenol-chloroform extraction and ethanol precipitation of DNA. Gapped or nicked DNA fragments were resolved on 6.5% denaturing polyacrylamide gels with 8 M urea including a sequencing ladder made from the same DNA. The relative abundance of DNA fragments of various sizes resulting from gaps or nicks in DNA from the mobile and immobile species were compared by analysis with ImageQuant and Microsoft Excel, and normalization of signals based on total DNA. Plots for the relative abundance of gaps or nicks in the mobile and immobile fractions were overlaid to identify nucleosomal regions where gaps or nicks interfere with remodelling.

**High-resolution mapping of changes in H2B53–DNA contacts.** Histone octamers containing cysteine at residue 53 of H2B were conjugated to APB immediately after octamer refolding. APB-modified octamers were reconstituted into 70N20 or 0N70 nucleosomes. ISW2 and INO80 were incubated with nucleosomes in conditions such that nucleosomes are completely bound at 30 °C for 15 min. For ISW2 remodelling, the samples were incubated at 18 °C for an additional 15 min. Nucleosome movement was analysed on 5% native polyacrylamide gels. For site-directed histone–DNA crosslinking samples were ultraviolet irradiated (3 min at 310 nm, 2.65 mW cm$^{-2}$). Samples were denatured with 0.1% SDS at 37 °C for 20 min in 30 mM NaCl and 20 mM Tris-HCl (pH 8.0). Crosslinked protein–DNA was enriched and separated from un-crosslinked DNA by phenol-chroloform (4:1) extraction; aqueous phase containing un-crosslinked DNA was discarded. Crosslinked DNA was ethanol precipitated with 1 M LiCl in presence of sheared salmon sperm DNA as carrier[57]. Crosslinked DNA was cleaved with 1 M pyrrolidine (Sigma) at 90 °C for 20 min. DNA samples were analysed on denaturing 6.5% polyacrylamide gels with 8 M urea along with a sequence ladder made from the same DNA, visualized by phosphorimaging and quantified by ImageQuant (Version 5.2). Total lane intensities were normalized for loading bias using Microsoft Excel.

**Nucleosome reconstitution for single-molecule FRET.** Recombinant *Xenopus laevis* histones were used for nucleosome reconstitution. The T112C residue of H2B was labelled with ATTO 647N or Cy5.5 via thiol-maleimide conjugation in the folded H2A.Z–H2B or H2A–H2B dimer. Nucleosome samples were prepared by combining H2A.Z–H2B or H2A–H2B dimer, (H3–H4)$_2$ tetramer and nucleosomal DNA in 2:1:1 stoichiometry and dialysing against 2,000, 850, 650, 500, 300 and 10 mM NaCl stepwise for 1 h in each step at 4 °C. Dimer labelling efficiency was adjusted to 50% by mixing labelled and unlabelled dimers. Nucleosomal DNA containing the Widom's 601 nucleotide positioning sequence[72] with a 70 bp dsDNA linker and a single-stranded 20 nt linker with a biotin labelled 5′-end was prepared by ligating seven or eight (for gap or nick at nt −56) oligonucleotide fragments (Supplementary Fig. 4). A Cy3 molecule replaced nt −37 on the lower strand.

**Exchange measurements with three-colour single-molecule FRET.** Three-colour single-molecule FRET measurements[74] on a total internal reflection fluorescence microscope were performed. Images were taken with iXon DU-897 camera (Andor Technology, Belfast, Ireland) at a 10 Hz frame rate with 100 ms integration. For the hybrid single-molecule assay implemented in this study,

nucleosomes containing ATTO 647N-labelled H2A.Z–H2B (1 nM) were incubated with INO80 (2 nM) and Cy5.5-labelled free H2A–H2B dimer (5 nM) in a buffer (20 mM Tris-HCl, pH 8.0) containing 5 mM $MgCl_2$, 1 mM ATP, 5% glycerol, 20 mM KCl, 50 mM NaCl, 0.05% Tween-20, 1 mM DTT, 0.1 mg $ml^{-1}$ BSA, 1 mM Trolox, 4 mM PCA, and 0.4 U $ml^{-1}$ PCD. After 30 min at 30 °C, the mixture was diluted 20-fold with the same buffer followed by injection into a sample chamber on a microscope slide. The fluorescence intensities from individual nucleosomes in the three spectral regions of 555–620, 655–690 and 690–750 nm were simultaneously recorded. Representative screen captures of a recorded movie are shown in Supplementary Fig. 5. Out of the particles showing acceptor photobleaching, a subset (typically 40–60%) that shows single-step acceptor and donor photobleaching with > 0.5 FRET efficiency were used for further analysis to filter out nucleosomes with two-labelled dimers, a labelled dimer on the opposite side of the DNA gyre containing Cy3, or no dimer. These criteria ensure that we analyse only the proximal-dimer-labelled nucleosome particles. The distal-dimer-labelled nucleosomes have much longer acceptor photobleaching lifetime, and therefore, an elongated movie recording time does not improve data collection efficiency. We then excluded particles that showed a signal-to-noise ratio < 3.0 (typically ∼ 40%) to avoid ambiguities in the analysis. These criteria are based solely on the signal quality, and therefore, do not bias the result. The three intensity time traces from a nucleosome allows us to distinguish whether the nucleosome contains H2A.Z–H2B or H2A–H2B. A nucleosome containing H2A.Z–H2B shows higher fluorescence intensity in the 655–690 nm region. A nucleosome containing H2A–H2B shows higher fluorescence intensity in the 690–750 nm region. The fraction of nucleosomes containing the exchanged dimer after 30 min incubation is presented in the results as the efficiency of dimer exchange by INO80.

**Nucleosome-binding assays.** INO80 (0, 2.5, 5, 10, 20 and 40 nM) was incubated with nucleosomes (40 nM) for 30 min at 30 °C in 10 mM Na-HEPES (pH7.8), 4 mM $MgCl_2$, 60 mM NaCl, 0.2 mM EGTA, 0.04 mM EDTA, 8% glycerol and 0.1 µg $µl^{-1}$ bovine serum albumin and analysed on 4% native polyacrylamide gels in 1 × Tris-EDTA buffer. INO80 (40 nM) was bound to nucleosomes (40 nM) in the presence of increasing amounts of sheared salmon sperm DNA as competitor, and incubated at 30 °C for 30 min. Reactions were analysed as for the nucleosome-binding assays.

**Kinetics of nucleosome remodelling using gel shift assays.** INO80 (26 nM) and ISW2 (57 nM) were pre-bound to 0N70 nucleosomes (25 nM) at 30 °C for 30 min. Reactions were stopped by adding ATP-γ-S and sonicated salmon sperm DNA (stop mix) to final concentrations of 1.5 mM and 300 ng $µl^{-1}$. Samples were analysed on 5% native polyacrylamide gels in 0.5 × TBE buffer. Error bars represent s.d. of the mean calculated from three technical replicates.

**Nucleosome-stimulated ATP hydrolysis.** ATPase assays were carried out with homogeneous reconstitutions, but under the same conditions as for remodelling assays. After pre-binding, γ-$^{32}$P-labelled ATP was added along with cold ATP. SDS and EDTA were added to final concentrations of 2% and 100 mM to stop reactions. The extent of ATP hydrolysis was determined by thin-layer chromatography by the amount of inorganic $^{32}$P (Pi) released[36].

**Statistical analysis.** For each independent *in vitro* experiment, at least three technical replicates were performed. Peptide mapping of Ino80–DNA interactions at nt − 58 was done at least three times each with two independent preparations of the INO80 complex, which showed identical results. Peptide mapping without ADP was done two times and showed results identical to that observed in the presence of ADP. Peptide mapping markers of known molecular weights were prepared two times and each set of markers were run on three separate 4–20% gradient SDS–polyacrylamide gels to determine $R_f$. Mapping histone–DNA contacts at residue 53 of histone H2B were done with three technical replicates for ISW2 at 16 µM ATP and once at 8 µM ATP. All replicates showed the same changes in contacts except that the replicate with the lower ATP concentration was slower. Similarly, two technical replicates with INO80 were done at 30 µM ATP and another at 800 µM ATP and gave similar results except that the higher ATP concentration was faster. Error bars represent s.e.m. calculated from three technical replicates. The sample sizes for the single-molecule FRET experiments are given in the tables, which are 643 and 734 for the two cases in Fig. 5b and 741, 760 and 502 for the three cases in Fig. 5c. The errors shown are the s.d.'s of the measurements assuming the Poissonian statistics. The results are from five or more independent measurements per case. Three different batches of nucleosome samples prepared on three different days were used in the measurements.

**Data availability.** The data sets generated and/or analysed during the current study are available from the corresponding authors on reasonable request.

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

## Acknowledgements

We acknowledge support provided by the National Institute of Health, R01 GM097286 (T.-H.L.) and R01 GM108908 (B.B.) and by the National Research Foundation of Korea NRF-2016R1D1A1B03933938 (J.K.). Also, we want to thank Briana Dennehey, Jim Persinger and members of the Bartholomew lab for critical reviewing of the manuscript.

## Author contributions

DNA footprint experiments were done by S.K.B. and M.I.U. DNA crosslinking experiments were done by S.B. and M.I.U. and peptide mapping by S.B. The DNA gap and nick interference experiments for INO80 were done by M.I.U. and for ISW2 by S.B. Mapping DNA movement near residue 53 of histone H2B was done by A.H. and S.B. and all single-molecule FRET experiments were done by J.K. Nucleosome mobilization using gel shift assays and ATP hydrolysis measurements were done by M.I.U. and S.G.H. The manuscript was written by S.B., M.I.U., T.-H.L. and B.B.; and the research directed by T.-H.L. and B.B. S.B., M.I.U. and J.K. all equally contributed to this paper.

## Additional information

**Competing interests:** The authors declare no competing financial interests.

