## [Peer Review File · Nature Communications]

Reviewers' comments:

Reviewer #1 (Remarks to the Author):

In this paper, Brahma and co-workers describe an interesting set of experiments aimed at understanding how INO80 translocates DNA and exchanges H2Az-H2B for H2A-H2B. Using hydroxyl radical footprinting, they show that unlike ISWI, INO80 contacts DNA further away from the dyad, at superhelical locations -3, -5 and -6. By transferring a radioactive label from a specific base on the DNA onto the protein and subsequently mapping the position of the label on the protein, they identify that the ATPase domain of INO80 (Ino80) contacts DNA around 58 nucleotides from the dyad, and that the interaction is independent of ATP. Gaps in the DNA that block translocation, hinder the ability of INO80 to mobilize nucleosomes, even when the gap is as far out as position -58 from the dyad. This is consistent with their model that INO80 mainly translocates DNA around the H2A-H2B dimer interface, away from the dyad region. A similar picture emerges in the case of nicks – nicks near the contact region hinder mobilization – an observation which the authors argue implies that torsional strain is required to initiate translocation. By chemically modifying histone H2B in order to crosslink DNA 54 nucleotides from the dyad, the authors visualize the translocation. They see a minor step of 11 nucleotides and a major step of 20 nucleotides, and an overall loss of contact with H2B on the side where translocation takes place. The long step size is consistent with INO80 acting at +1 nucleosomes, drawing DNA in from the NFR. Using single molecule fluorescence to monitor the replacement of H2AZ with H2A and vice versa, the authors conclude that it is 10 times more likely for H2AZ to be replaced with H2A than the reverse, and that this bias is lost when translocation or torque buildup is prevented either by introducing gaps in the DNA or nicks.

These experiments are suggestive of a novel mechanism employed by INO80 where translocation is effected away from the dyad, but again, the evidence is indirect as footprinting experiments measure ensembles of configurations and bulk averages. The single molecule experiments are also at best indirect, as there is no direct simultaneous measure of DNA translocation/ torsion coupled to H2A exchange. While this remains a good piece of work, I feel it may be suitable for a more specialized journal.

Reviewer #3 (Remarks to the Author):

In this manuscript, the authors investigated the mechanism of exchange of nucleosomal H2A.Z for H2A catalyzed by chromatin remodeler INO80 using hydroxyl radical footprinting, site-specific cross-linking, single nucleotide gaps and nicks, and single molecule FRET. They found that INO80 exchanges H2A.Z for H2A through translocating on DNA-proximal to histone dimers.

This is a very detailed study on the catalytic mechanism of INO80 remodeler. The results are new and should be interesting to people in the chromatin field. Overall, it is suitable for publication in Nature communication.

Other comments:

(1) The authors used an asymmetric nucleosome with flanking DNA on one side of the nucleosome in their study and found that INO80 exchanges one H2A.Z for H2A through translocating on DNA-proximal to histone dimers. In vivo, it is likely that there are two flanking DNA for one nucleosome. It is not clear to me why the authors chose the asymmetric nucleosome, in particular, the catalytic reaction was found to involve DNA translocation. One wonders what would happen if there are flanking

DNA on both sides of the nucleosome. Could both H2A.Z molecules in the nucleosome be exchanged for H2A?

(2) The cryo-EM structure of INO80 has been determined at low resolution and the nucleosome binding location has been suggested. The relationship between the cryo-EM structure and the results presented here should be discussed.

(3) It would be helpful for general readers to follow their work if the authors could introduce the principle of DNA gap and nick experiments.

Response to Reviewers' comments:

Reviewer #1:

1. These experiments are suggestive of a novel mechanism employed by INO80 where translocation is effected away from the dyad, but again, the evidence is indirect as footprinting experiments measure ensembles of configurations and bulk averages.

The experimental evidence provided in our paper for INO80 translocating on nucleosomal DNA near the edges of nucleosomes rather than at the dyad axis is based on several observations and not solely on DNA footprinting. Not only do we identify where INO80 binds to nucleosomal and extranucleosomal DNA by DNA footprinting, but we show by site-directed DNA crosslinking and peptide mapping that the ATPase domain of INO80 contacts DNA 58 base pairs from the dyad axis. At this location the ATPase domain is associated near the H2A-H2B dimer and is only 14-15 bps from where DNA enters into nucleosomes. Besides showing that the DNA translocation domain is physically bound at the edges of nucleosomes, we confirm INO80 functions in this manner through site-specific interference experiments. For these experiments we took advantage in two distinct ways of the fact that gaps in DNA block translocation of these type of enzymes. First, we use DNA gap interference in an unbiased screen to scan for the positions where gaps block DNA translocation by INO80 immediately after addition of ATP. Second, we start to INO80 translocating along DNA from a distal location to a site proximal to the active site. The site where it eventually stops is indicative of the location of the active site of the enzyme in terms of a nucleosomal position. Both of these approaches confirm that enzyme translocation is blocked when it encounters a gap ~58 nts from the dyad axis, perfectly in agreement with the physical data of the ATPase domain being bound at this position.

As to the potential difficulties of bulk average and conformational heterogeneity that can cause problems with ensemble techniques, we have found INO80-nucleosome complexes to be uniform and homogeneous as visualized by a single band after separation on a native

polyacrylamide gel. Also if there were significant heterogeneity there would be partial footprints instead of the complete protections on the nucleosome that we observe. The footprints we observed are strongly indicative of a fairly homogeneous set of INO80-nucleosome complexes.

2. The single molecule experiments are also at best indirect, as there is no direct simultaneous measure of DNA translocation/ torsion coupled to H2A exchange.

Not only do we use ensemble based techniques, but also single molecule techniques to examine the effects of DNA gaps for blocking INO80 translocation and dimer exchange. The only reasonable conclusion we can arrive at with a high degree of certainty using the combined sets of data is that INO80 translocates at the edge of nucleosome near the H2A-H2B dimer and promotes dimer exchange. We are

not able to show a real-time movie of translocation and simultaneous dimer exchange, which for all practical purposes is not technically feasible at this time for us or anyone else because of the extremely limited photostabilities of the fluorophores that are commercially available. However, we are the first to use a three color single molecule FRET based assay to examine dimer exchange by a chromatin remodeler and the first to show in this manner the preference of INO80 for replacing H2A.Z dimers in nucleosomes for H2A dimers. We are also the first to show that blocking DNA translocation interferes with the transfer of dimers. And last of all we show for the first time that INO80 stably displaces DNA from the H2A-H2B interface using ensemble based techniques and this displacement only occurs in the region where DNA translocation occurs and not on the other symmetrical position of the nucleosome.

Reviewer #3

1. This is a very detailed study on the catalytic mechanism of INO80 remodeler. The results are new and should be interesting to people in the chromatin field. Overall, it is suitable for publication in Nature communication.

We agree with and are grateful the reviewer appreciates the level of detailed analysis of the mechanism of INO80.

2. The authors used an asymmetric nucleosome with flanking DNA on one side of the nucleosome in their study and found that INO80 exchanges one H2A.Z for H2A through translocating on DNA-proximal to histone dimers. In vivo, it is likely that there are two flanking DNA for one nucleosome. It is not clear to me why the authors chose the asymmetric nucleosome, in particular, the catalytic reaction was found to involve DNA translocation. One wonders what would happen if there are flanking DNA on both sides of the nucleosome. Could both H2A.Z molecules in the nucleosome be exchanged for H2A?

In our earlier studies with INO80, we found that INO80 required 70 bp of linker DNA for optimal binding and mobilization of nucleosomes. Making asymmetric nucleosomes constrains INO80 to bind in one preferred orientation rather than having the option of binding in one of two equally likely orientations if there were 70 bp of linker DNA at both ends of nucleosomes. As to whether INO80 could exchange out both copies of H2A.Z, based on other's experiments

especially those with SWR1 we expect that the enzyme would not exchange both simultaneously but would first exchange out one dimer followed by exchange of the second dimer. We however are not examining the sequence of these type of events in this study.

3. The cryo-EM structure of INO80 has been determined at low resolution and the nucleosome binding location has been suggested. The relationship between the cryo-EM structure and the results presented here should be discussed.

The CX-MS and cryo-EM data from the Hopfner group (Tosi et al. Cell 2013) finds the ATPase domain of INO80 interacts close to the H2A-H2B dimer and near the DNA entry site of nucleosomes as shown by the RecA2 lobe of INO80 crosslinking the L2 loop region of histone H2A and the insertion near RecA2 crosslinking to the N-terminus of H3. These data coincide remarkably well with where we found the ATPase domain of INO80 bound to nucleosomes.

4. It would be helpful for general readers to follow their work if the authors could introduce the principle of DNA gap and nick experiments.

We have revised our text to better provide information regarding the principle behind DNA gaps and nick interfering with DNA translocation and chromatin remodeling.

Reviewers' comments:

Reviewer #1 (Remarks to the Author):

The manuscript has improved in clarity and presentation, and the authors certainly have obtained a body of interesting data that distinguishes INO80 from other remodellers. I am still not convinced by their claim that INO80 selectively removes H2A.Z because the only piece of data supporting that claim is single molecule data which has its own issues (see below). In addition, according to Reference 39, "H2A.Z eviction was unaffected upon depletion of INO80, a remodeler previously reported to displace nucleosomal H2A.Z.". How do the authors reconcile their claim with Reference 39? Reference 39 was discussed in the context of nucleosome mobilization in the discussion section but not in the context of H2A.Z eviction.

About single molecule data: My main problem with experiment is that the authors do not show any primary data, for example two or three color images showing visually how robust the results are before manually selecting 'good' traces. The readers do not get any idea on what fraction of fluorescent spots are rejected from analysis. The other problem is that the experiment shows only one run of positive experiment which showed a fair low yield (less than 10%) of H2A.Z exchange. All others are negative experiments. If the experiment is difficult to perform reproducibly and only a small fraction of runs works, negative controls are not very meaningful. They should report at least three independent experiments of each kind.

The authors' response to Reviewer 3's comments on the cryoEM structure by Hopfner (Cell 2013) is not satisfactory. First, the paper is not on the reference list. Second, they should try to put their nucleosome/INO80 contact information in the structural model of the complex to see if the overall arrangement is consistent with the structural data.

"and now for the first time we have found one that utilizes the flexibility at SHL5 to mobilize nucleosomes and exchange histone dimers." This sentence is unclear. Ngo et al (Cell 2015) related DNA local flexibility to nucleosome stability under tension. Are the authors referring to flexibility of similar type?

Reviewer #3 (Remarks to the Author):

My questions have been addressed in the revised manuscript. I am ok for its publication.

Response to Reviewer's comments:

Q: The manuscript has improved in clarity and presentation, and the authors certainly have obtained a body of interesting data that distinguishes INO80 from other remodelers. I am still not convinced by their claim that INO80 selectively removes H2A.Z because the only piece of data supporting that claim is single molecule data which has its own issues (see below). In addition, according to Reference 39, "H2A.Z eviction was unaffected upon depletion of INO80, a remodeler previously reported to displace nucleosomal H2A.Z.". How do the authors reconcile their claim with Reference 39? Reference 39 was discussed in the context of nucleosome mobilization in the discussion section but not in the context of H2A.Z eviction.

R: *We actually did attempt to reconcile the authors' data from reference 39 with ours in the discussion towards the bottom of page 17. In reference 39 they are merely examining the role of INO80 at promoters, but INO80 targets many more sites than these such as in DNA damage repair. While INO80 may not be the major H2A.Z exchanger at promoters, it may have a much more prominent role at other genomic sites.*

The pertinent paragraph now reads as follows:

"The precision of the dimer exchange mechanism of INO80 may make the removal of H2A.Z by INO80 less robust than the elongating RNA polymerase II complex that can completely remove H2A.Z nucleosomes including the H3-H4 tetramer³⁹. While INO80 may not be the primary or only complex to remove H2A.Z at promoters³⁹, it may be a major factor for removing H2A.Z at DNA damage sites or at centromeres and telomeres. Our observation of the less efficient mobilization of nucleosomes by INO80 compared to ISW2 is also consistent with in vivo mapping of nucleosome positioning. Reduction of INO80 does not have as significant effect on nucleosome positioning as does loss of ISW2 or ISW1 and yet causes nucleosomes to be more "fuzzy"³⁹, consistent with the lowered efficiency of INO80 to mobilize nucleosomes."

Q: About single molecule data: My main problem with experiment is that the authors do not show any primary data, for example two or three color images showing visually how robust the results are before manually selecting 'good' traces. The readers do not get any idea on what fraction of fluorescent spots are rejected from analysis. The other problem is that the experiment shows only one run of positive experiment which showed a fair low yield (less than 10%) of H2A.Z exchange. All others are negative experiments. If the experiment is difficult to perform reproducibly and only a small fraction of runs works, negative controls are not very meaningful. They should report at least three independent experiments of each kind.

R: In regard to the reproducibility of our experiments, we have done multiple independent measurements that are clearly stated in Figure 5. To reiterate, our results are based on at least 5 independent sets of measurements from 3 different batches of nucleosome samples that were prepared on three different days. The reviewer is concerned about an apparent lack of “primary” data, but we provide single fluorophore intensity traces, which are well accepted as “primary” data. You will rarely see a screen shot of single molecule experiment in a regular biochemistry paper because it is a very inefficient way of communicating the data. Regardless, we have provided typical screen captures of our measurements in the revision as requested (Fig. S5).

The reviewer is concerned about the exchange rate being low at 10%, but this is not at all low. This efficiency is comparable to, if not higher than, the published biochemical result (Cell (2011) **144** 200), which was 35% for 60 min incubation at two orders of magnitude higher nucleosome/dimer concentrations and a 2.5-fold higher INO80 concentration than ours. We cannot compare the nominal values of the efficiency side-by-side because the kinetic elementary steps are unknown. Regardless, 10 % is fairly efficient for such a complex enzymatic reaction in vitro at nM concentrations of the substrate and enzyme for 30 min. The reason why we used lower concentrations and a shorter reaction time is because these are the conditions that balance well between reaction efficiency and signal quality. At a higher substrate concentration, non-specific binding becomes non-negligible. During longer incubation, nucleosomes start partially disassembling. These two effects would have dramatically complicated the data analysis. At an ensemble level, partial disassembly of the nucleosomes would not even be resolved and the error will be convolved in the result. All in all, our single molecule results demonstrate the enzymatic function of INO80 more clearly than the biochemical result.

The reviewer may not understand why single-molecule data filtering that is based solely on signal quality would not bias the result. Our filtering is based solely on the signal quality, which has nothing to do with the molecular function. The reason why we filtered the signal quality is to avoid ambiguities in the analysis, in order to minimize the uncertainty. We select the nucleosomes displaying low-noise, unambiguous fluorescence time trajectories with single-step photobleaching events for both the donor and the acceptor with a high FRET. Such nucleosomes constitute on average 16% of the total nucleosomes observed. This is 64 % of the nucleosomes that are properly labeled for the measurement (i.e. 16 % = 64 % of the 25 % that has only the proximal-dimer labeled according to the 1:1 labeled:unlabeled dimer ratio in the sample). These signal-quality filtering criteria were applied to all of the samples equally in all of the cases presented, and thus cannot bias the results, and this description has been added (Fig. S5).

We should also point out that we show our single molecule FRET data as is generally accepted in the field with a sample trace, a sample image and a count histogram.

Q: The authors' response to Reviewer 3's comments on the cryoEM structure by Hopfner (Cell 2013) is not satisfactory. First, the paper is not on the reference list. Second, they should try to put their nucleosome/INO80 contact information in the structural model of the complex to see if the overall arrangement is consistent with the structural data.

R: *We have corrected the oversight of not listing the Hopfner (Cell 2013) paper in the reference list even though it was cited within the text. We have added text that compares and contrasts our findings to the observations from the Hopfner group. The RecA2 lobe of Ino80 crosslinks to the L2 loop region of histone H2A, and the insertion near the RecA2 lobe crosslinks to the N-terminus of H3. Therefore the CX-MS data places the ATPase domain of Ino80 close to the SHL5 and 6 positions in nucleosomes. The cryo-EM of INO80 shows an elongated complex with two large lobes interconnected by a hinge region and the ATPase domain of Ino80 is located at the hinge. The Hopfner model based on both cryoEM and CX-MS has nucleosomes bound near the hinge region of INO80 with the ATPase domain close to the SHL5/6 positions in nucleosomes. These data coincide remarkably well with where we find the ATPase domain of INO80 bound to nucleosomes.*

Q: "and now for the first time we have found one that utilizes the flexibility at SHL5 to mobilize nucleosomes and exchange histone dimers." This sentence is unclear. Ngo et al (Cell 2015) related DNA local flexibility to nucleosome stability under tension. Are the authors referring to flexibility of similar type?

R: *We are not referring to DNA flexibility towards bending as in Ngo et al. (Cell 2015), which lends to enhanced binding of nucleosomal DNA to the histone octamer. Rather we are referring to the DNA twisting flexibility that has been observed in the crystal structure of nucleosomes in which DNA deviates from normal B-DNA with slight overtwisting or underwinding of DNA at this position. This kind of distortion is likely to occur when chromatin remodeling occurs and is thus pertinent to our discussion. We have added text to make this clearer to the reader.*

REVIEWERS' COMMENTS:

Reviewer #1 (Remarks to the Author):

I thank the authors for kindly answering my questions clearly. The only remaining concern I have is about the new Figure S5. It does not show any visible evidence of histone exchange. There are a few spots in the H2A channel after the reaction but these spots are accompanied by even brighter spots in the H2A.Z channel and cannot be due to exchange. Why? If they are showing the images from the negative control, I suggest they replace them with images from the exchange experiment that showed 10% reaction yield. In addition, the number of H2A.Z spots decrease dramatically after the reaction, which suggests that INO80 evicts H2A.Z. I don't think that this much bigger effect is mentioned in the paper.

Response to Reviewers' comments:

Q: The only remaining concern I have is about the new Figure S5. It does not show any visible evidence of histone exchange. There are a few spots in the H2A channel after the reaction but these spots are accompanied by even brighter spots in the H2A.Z channel and cannot be due to exchange. Why? If they are showing the images from the negative control, I suggest they replace them with images from the exchange experiment that showed 10% reaction yield. In addition, the number of H2A.Z spots decrease dramatically after the reaction, which suggests that INO80 evicts H2A.Z. I don't think that this much bigger effect is mentioned in the paper.

A: We addressed the comment by revising figure S5 and providing more detailed information on how to read the figure in the caption. The reviewer mistook that the starting image (Fig. S5A) is before INO80 incubation and the final image (Fig. S5B) is after incubation, which is not the case. The reaction had already taken place in an eppendorf tube before we started imaging. We needed to take a movie of the particles in order to select properly labeled nucleosomes that show single-step photobleaching of fluorophores (time trajectories in Fig. 5). In the revised figure S5, we show and explain clearly how the H2A.Z retained and replaced nucleosomes look before and after photobleaching and demonstrate how the measurement was made.

In figure S5A, we identified total 26 properly assembled nucleosomes out of which 3 show a sign of H2A.Z replaced by H2A ($= 3/26 = 10 \pm 6\%$ exchange efficiency, note single significant figure, error is based on the standard binomial distribution). In the final image (Fig. S5B), 17 out of the 26 nucleosomes have both donor and acceptor photobleached that include two H2A nucleosomes ($= 2/17 = 10 \pm 7\%$ exchange efficiency, note single significant figure, error is based on the standard binomial distribution). The results given in the paper (Fig. 5) are from many repeated measurements ($n > 500$, Fig. 5).